# Data-Adaptive Relaxed Equivariant Networks for Symmetry Breaking

**Yuxuan Chen**
Khoury College of Computer Sciences
Northeastern University
chen.yuxuan7@northeastern.edu

**Robin Walters**
Khoury College of Computer Sciences
Northeastern University
r.walters@northeastern.edu

## Abstract

Equivariant neural networks improve generalization by incorporating symmetry, but real-world data often breaks symmetry in complex ways. In particular, different orbits in the data may follow different symmetry patterns or degrees of breaking. Existing relaxed equivariant models use shared weights across all orbits, which is less effective in such heterogeneous settings. We propose Data-Adaptive Relaxed Equivariant Networks (DAREN), a method that learns to adjust symmetry behavior separately for each orbit. By generating orbit-specific relaxed weights and using a gating mechanism to control symmetry strength, our model adapts to varying symmetry level across the data. Experiments on synthetic datasets show that DAREN outperforms equivariant models, relaxed models and unconstrained models, especially when symmetry varies across regions.

## 1 Introduction

Encoding symmetry as an inductive bias into neural networks has proven effective for improving efficiency and generalization across domains such as vision (Weiler et al., 2018; Deng et al., 2021; He et al., 2021; 2022; Li et al., 2024), sciences (Wang et al., 2021; Liao and Smidt, 2022), and robotics (Huang et al., 2022; Zhu et al., 2025; Hu et al., 2025). Group equivariant networks like G-CNNs (Cohen and Welling, 2016) enforce strict symmetry constraints, but they can underperform when data exhibit only partial or approximate symmetry due to factors like noise or intrinsic symmetry breaking (Wang et al., 2022). To address this, recent works (Romero and Lohit, 2022; van der Ouderaa et al., 2022; Hofgard et al., 2024; Park et al., 2024; Wang et al., 2022; 2024b) propose relaxed equivariant models that incorporate learnable symmetry relaxation. Notably, Relaxed Group Convolution (Wang et al., 2022; 2024b) introduces learnable weights over the group domain to enable partial symmetry modeling. While effective under global symmetry breaking, for domains which have varying levels of symmetry across the domain, relaxed equivariant models must always choose the lowest level of symmetry, reducing their effectiveness.

We propose Data-Adaptive Relaxed Equivariant Networks (DAREN) in which the degree of symmetry breaking is conditioned on the input and thus can learn which parts of the domain exhibit symmetry break. In particular, we circumvent the limitations of the global relaxation in Wang et al. (2024b) by introducing the notion of *orbitwise equivariance*. This allows the model to divide up the domain by equivalence classes determined by the group action and learn different levels of symmetry for each. In real-world scenarios, some orbits may exhibit perfect equivariance while others may demonstrate substantial symmetry violations. Applying a shared symmetry relaxation across all orbits imposes a tradeoff—one that can hurt performance on fully equivariant regions, especially under large group actions where strict equivariant model is known to be most effective.

In contrast, DAREN learns to adapt the model's equivariance behavior to the symmetry structure of each individual orbit. Our approach is based on the following two mechanisms:

- **Adaptive Relaxation**: We introduce a generator network that maps each orbit (represented by an element in the quotient space $X/G$) to a set of orbit-conditioned relaxed weights, allowing the model to capture orbit-specific symmetry profiles.

- **Gated Control**: To further improve robustness and stabilize training, we implement a gating mechanism that dynamically interpolates between original and averaged weights over the group.

Our model retains the benefits of relaxed equivariance, while introducing the capacity to respond to local symmetry variations in a principled and learnable manner. Our contributions can be summarized as follows:

- We introduce the concept of *orbitwise equivariance* to characterize the symmetry behavior of functions with respect to group orbits, enabling a more nuanced understanding of equivariance in the context of symmetry breaking.
- We propose Data-Adaptive Relaxed Equivariant Networks (DAREN) that learn to adapt the model's equivariance behavior to the symmetry structure of each individual orbit, allowing for more effective modeling of symmetry breaking.
- Experiments on synthetic data shows that models of DAREN outperforms both strictly equivariant models, relaxed models and unconstrained model, especially when symmetry varies across regions.

## 2  RELATED WORKS

**Symmetry breaking.**  Symmetry breaking is a fundamental phenomenon in many real-world systems, where the underlying symmetries of the system are not perfectly preserved due to factors such as noise or intrinsic properties of the system (Lee and Yang, 1956; Anderson, 1972). Here we mainly focus on the works that study symmetry breaking in the context of equivariant neural networks. Smidt et al. (2021) study symmetry breaking from a data-centric perspective, showing that strictly equivariant Euclidean neural networks can be used to identify missing symmetry-breaking order parameters implied by the dataset via gradient-based analysis. Lawrence et al. (2025) reinterpret symmetry breaking in equivariant neural networks from a probabilistic perspective, showing that randomized canonicalization enables equivariant models to sample symmetry-broken outputs while remaining equivariant at the distribution level, and instantiate this idea via symmetry-breaking positional encodings. Lawrence et al. (2026) also study symmetry breaking from a distributional perspective, and propose a two-sample classifier test to quantify the extent to which a dataset violates an assumed symmetry through data augmentation. Wang et al. (2023) show that equivariant models can remain highly effective even when the imposed symmetry does not exactly match the latent symmetry of the task, distinguishing correct, incorrect, and extrinsic equivariance and demonstrating that extrinsic equivariance can still provide a useful inductive bias. Wang et al. (2024a) further formalize this perspective and develop a general theory of symmetry mismatch, introducing pointwise correct, incorrect, and extrinsic equivariance and deriving error lower bounds that characterize when such mismatches are beneficial or fundamentally harmful. Following this framework, our work formalizes symmetry mismatch at the level of group orbits, explicitly modeling orbit-wise correct and incorrect equivariance rather than assuming the equivariance type over the entire domain or pointwise.

**Relaxed equivariant networks.**  To handle symmetry breaking in data, several works have proposed relaxed equivariant models that allow for soft or partial symmetry. Romero and Lohit (2022) propose Partial G-CNNs, which learn layer-wise partial equivariance by adapting the effective subset of group transformations through learned sampling distributions. van der Ouderaa et al. (2022) introduce non-stationary continuous filters that relax strict equivariance by allowing convolutional kernels to depend on absolute group elements, enabling a continuous interpolation between equivariant and non-equivariant mappings at the layer level. Wang et al. (2022) propose relaxed group and steerable convolutions by introducing learnable relaxed weights indexed by absolute group position, providing a lightweight extension that relaxes strict equivariance with minimal additional parameters. Wang et al. (2024b) further extend this framework to systematically identify and interpret symmetry breaking in physical systems by analyzing the learned relaxed weights. Park et al. (2024) further apply relaxed group convolution in reinforcement learning by formalizing approximately equivariant MDPs, enabling models to handle symmetry breaking in both transitions and rewards. Since relaxed group convolution enables symmetry breaking through minimal modifications to standard equivariant

architectures, our DAREN framework is built upon this line of work and extends it by adapting the degree of equivariance in a data-dependent, orbit-wise manner.

## 3   ORBITWISE EQUIVARIANCE TYPE

We first adapt the definition of pointwise equivariance by Wang et al. (2024a) to setting on pairs $(x, g)$ and ground true functions, and then introduce the notion of orbitwise equivariance to capture the symmetry behavior of functions with respect to group orbits. Let $f \colon X \to Y$ be a task function where the group $G$ acts on the input space $X$ and output space $Y$. The group action on $X$ naturally partitions the domain into *orbits* $Gx = \{gx : g \in G\}$. The set of orbits is denoted $X/G$. In this paper, we restrict our attention to settings where the probability density is strictly positive over the entire domain, i.e., $p(x) > 0, \forall x \in X$, and consequently $p(gx) > 0, \forall g \in G$. In this setting, focusing on pair $(x, g)$ and a function $f$, we modified pointwise correct equivariance and pointwise incorrect equivariance definitions from Wang et al. (2024a) to the following:

**Definition 1** (Pointwise Correct Equivariance). *The pair $(x, g)$ has correct equivariance in function $f$ if $f(gx) = gf(x)$.*

**Definition 2** (Pointwise Incorrect Equivariance). *The pair $(x, g)$ has incorrect equivariance in function $f$ if $f(gx) \neq gf(x)$.*

While pointwise definitions describe local equivariance behavior, they fail to reflect the global group structure induced by group actions, which naturally operate over entire orbits rather than isolated points. Since equivariance is inherently defined with respect to a group $G$, it is natural to study the symmetry behavior not only at individual points but across the entire group orbit of a point.

To this end, we introduce the notion of *orbitwise equivariance*, which considers whether the equivariance condition holds consistently across all points in a given orbit $Gx$. This concept is particularly useful for identifying coherent patterns of symmetry or symmetry breaking in the function $f$, and forms the foundation for the adaptive equivariant mechanisms proposed in the following section.

**Definition 3** (Orbitwise Correct Equivariance). *We say that the orbit $Gx$ has correct equivariance in the function $f$ if for any $\hat{x} \in Gx$, and for all $g \in G$, we have $f(g\hat{x}) = gf(\hat{x})$.*

**Definition 4** (Orbitwise Incorrect Equivariance). *We say that the orbit $Gx$ has incorrect equivariance in the function $f$ if there exists $\hat{x} \in Gx$ and $g \in G$ such that $f(g\hat{x}) \neq gf(\hat{x})$.*

It is clear that if an orbit has correct equivariance, then all points on the orbit satisfy the pointwise correct equivariance condition. If an orbit has incorrect equivariance, it may contain points that satisfy the pointwise correct equivariance condition, but at least one point in the orbit must violate it. The symmetry breaking degree of an orbit can be captured by equivariance error (EE), which is computed as follows:

$$\mathrm{EE}(Gx, f) = \mathbb{E}_{g \in G} \left[ \| f(g\hat{x}) - gf(\hat{x}) \| \right]. \tag{1}$$

We denote $\mathrm{EE}(Gx, f)$ for the orbit $Gx$ instead of the element $\hat{x}$ since the definition does not depend on the specific choice of $\hat{x} \in Gx$. This value quantifies the degree of symmetry breaking within the orbit, with higher values indicating stronger violations of equivariance. This orbitwise perspective allows us to move beyond localized mismatches and instead focus on global symmetry consistency across orbits. In this paper, we mainly consider datasets exhibiting heterogeneous orbitwise symmetry breaking, where each orbit may vary in both equivariance validity (correct vs. incorrect) and the degree of symmetry violation (as quantified by the equivariance error). In the following sections, we propose a data-adaptive framework that automatically learns the symmetry behavior of each orbit from data, without requiring prior knowledge of equivariance types.

## 4   DATA-ADAPTIVE RELAXED EQUIVARIANT NETWORKS

To address the challenge of heterogeneous orbitwise symmetry breaking, we propose a novel architecture called **Data-Adaptive Relaxed Equivariant Networks (DAREN)**. Unlike conventional equivariant networks that enforce uniform group symmetry across the entire domain, or unconstrained neural networks that learn the symmetry of the data without any inductive bias, starting from an equivariant initialization DAREN can automatically learn and adapt to the symmetry breaking degree

of each orbit presented in the data. We begin by reviewing the Relaxed Group Equivariant Network (Wang et al., 2022; 2024b), which serves as the foundation of our approach and provides a flexible mechanism for modeling global symmetry breaking.

## 4.1 Relaxed Group Equivariant Networks

For simplicity, we focus only on the group structure and consider Relaxed Group Equivariant MLP cases, which can be seen as $1 \times 1$ convolutional layers. The form is easily extendable to group convolutional layers. Let $f_0 \in \mathbb{R}^{C_0}$ be the input feature with defined group transformation $\pi_g[f_0]$, where $\pi_g$ is the group action on the input feature, and $f_1 \in \mathbb{R}^{C_1 \times |G|}$ be the output feature which can be seen as $C_1$ dimensional features defined on group space of $G$. We use the notation $(f_1)_g$ to denote the $g$-th group channel of the output feature $f_1$. Let $\{w_g^{(l)} \in \mathbb{R} : g \in G, 1 \le l \le L\}$ be the trainable parameters where $L$ is the number of filter banks, and $\{\psi_{g^{-1}g'}^{(l)} \in \mathbb{R}^{C_1 \times C_0} : g \in G, 1 \le l \le L\}$ be the trainable filters. The Relaxed Lifting Layer can be defined as follows:

$$(f_1)_g = (f_0 \widetilde{\star} \psi)_g = \sum_{g' \in G} \sum_{l=1}^{L} w_g^{(l)} \psi_{g^{-1}g'}^{(l)} \cdot \pi_{g'}[f_0], \tag{2}$$

Similarly, the middle layer of Relaxed Group Equivariant MLP can be defined as:

$$(f_2)_g = \sum_{g' \in G} \sum_{l=1}^{L} w_g^{(l)} \Psi_{g^{-1}g'}^{(l)} \cdot (f_1)_{g'}, \tag{3}$$

We can see that the learnable weights $w_g^{(l)}$ can be different across group dimension, which allows the model to break the weight-sharing constraint of equivariant networks.

## 4.2 Limitations of Relaxed Equivariant Networks

Although Relaxed Group Equivariant Networks provide a flexible framework for modeling symmetry breaking, the learnable weights $w_g^{(l)}$ are shared across all orbits. However, this is a limitation when the degree of symmetry varies across orbits. For example, consider the case of two orbits $Gx_1$ and $Gx_2$, where $Gx_1$ exhibits correct equivariance and $Gx_2$ exhibits incorrect equivariance. When Relaxed Group Equivariant Networks are trained on the mixed data, however, the network inevitably learns a compromise between orbits with correct and incorrect equivariance.

This tradeoff may lead to suboptimal performance on correct equivariance orbits, especially under large group actions where even slight symmetry breaking can significantly degrade generalization. (see Section.5 for experiments). In addition, if the degree of symmetry breaking varies significantly across orbits, during training, the loss gradients at each orbits will be globally pooled to train the shared relaxed weights $w_g^{(l)}$, which may not accurately reflect the local symmetry breaking degree of each orbit.

## 4.3 Data-Adaptive Relaxed Weights

To overcome these limitations, we design a conditioning mechanism that generates orbit-specific relaxed weights, thereby enabling the model to adapt its symmetry behavior to match the properties of each orbit. By conditioning the weight generator on the quotient space $X/G$, we encode the orbit's geometric identity as an inductive bias, enabling the network to modulate its symmetry adaptively across heterogeneous regions of the input space.

To achieve this, we replace the single set of relaxed weights $w_g^{(l)}$ with a learnable function $\mathbf{w} : X/G \to \mathbb{R}^{|G|}$ that maps each orbit in the quotient space to a potentially different set of relaxed weights. We implement this function as a neural network that takes the orbit's geometric identity as input and outputs the corresponding relaxed weights. We refer to this approach as **Adaptive Relaxed Model (ARM)** in the following content. The Relaxed Lifting layer and Relaxed Equivariant layer of

ARM can then be modified as follows:

$$(f_1)_g = (f_0 \widetilde{\star} \psi)_g = \sum_{g' \in G} \sum_{l=1}^{L} \mathbf{w}_g^{(l)}(Gx) \cdot \psi_{g^{-1}g'}^{(l)} \cdot \pi_{g'}[f_0], \tag{4}$$

$$(f_2)_g = \sum_{g' \in G} \sum_{l=1}^{L} \mathbf{w}_g^{(l)}(Gx) \cdot \Psi_{g^{-1}g'}^{(l)} \cdot (f_1)_{g'}, \tag{5}$$

where $\mathbf{w}_g^{(l)}(Gx)$ for $Gx \in X/G$ is the relaxed weight generated by the function $\mathbf{w}$ at element $g$ at $l$-th filter bank. To ensure the equivariant initialization, we initialize the weight and bias of the last layer of the generator $\mathbf{w}$ to be zero. This approach allows the model to learn orbit-specific relaxed weights directly from the data, without requiring prior knowledge of the equivariance types or symmetry breaking degree of each orbit.

## 4.4 GATED CONTROL

However, when the the group size gets larger, directly learning the relaxed weights from the data can sometimes be challenging. In this case, for the orbits with correct equivariance, it is hard to learn an invariant function that can generate the same relaxed weights for all points in the orbit. To address this issue, we introduce a gate mechanism that adaptively controls the degree of model equivariance. The gate mechanism is implemented as a learnable function $a : X/G \to [0, 1]$ that outputs a scalar value for each orbit in the quotient space. The gated mechanism can be directly applied to Relaxed Group Equivariant Networks cases, enabling the model to learn the symmetry breaking degree of each orbit. We refer to this approach as **Gated Relax Model (GRM)** in the following content. We enable the generated gated value $a \in [0, 1]$ to control how much symmetry is enforced by interpolating between the original relaxed weights and their group-wise average. The generated relaxed weights of GRM can be computed as follows:

$$\mathbf{w}_g^{(l)}(Gx) = a(Gx) \cdot w_g^{(l)} + (1 - a(Gx)) \cdot \frac{1}{|G|} \sum_{g' \in G} w_{g'}^{(l)}, \tag{6}$$

where $a(Gx)$ is the gate value for the orbit $Gx$.

Combine the gated mechanism with the adaptive relaxed weights, which the relaxed weights are generated by the function $\mathbf{w}$, and the gate value is generated by the function $a$, we can obtain a new model named as **Gated Adaptive Relaxed Model (GARM)**. The relaxed weights of GARM can be computed as follows:

$$\hat{\mathbf{w}}_g^{(l)}(Gx) = a(Gx) \cdot \mathbf{w}_g^{(l)}(Gx) + (1 - a(Gx)) \cdot \frac{1}{|G|} \sum_{g' \in G} \mathbf{w}_{g'}^{(l)}(Gx), \tag{7}$$

Note that in some situations, it is not easy to explicitly express the quotient space $X/G$. In this case, we can use an invariant network as the backbone for the adaptive and gated mechanism.

## 5 EXPERIMENTS

In this section, we evaluate the effectiveness of our proposed Data-Adaptive Relaxed Equivariant Networks (DAREN) on synthetic datasets that exhibit heterogeneous symmetry breaking across different regions of the input space.

## 5.1 HETEROGENEOUS SWISS ROLLS

**Dataset.** The separated Swiss Roll dataset was proposed by Wang et al. (2024a) to study the behavior of equivariant networks under different equivariance types. The seperated swiss rolls consists of two interleaved 2D spirals with a distinct $z$ value to separate them vertically. By varying the label assignments and the $z$-level distributions of the spirals, the dataset enables construction of data that exhibits incorrect or correct equivariance relative to a $C_2$-invariant model. The group transformation on the swiss rolls data is $g(x, y, z) = (x, y, 1 - z)$, so the quotient space $X/G$ is

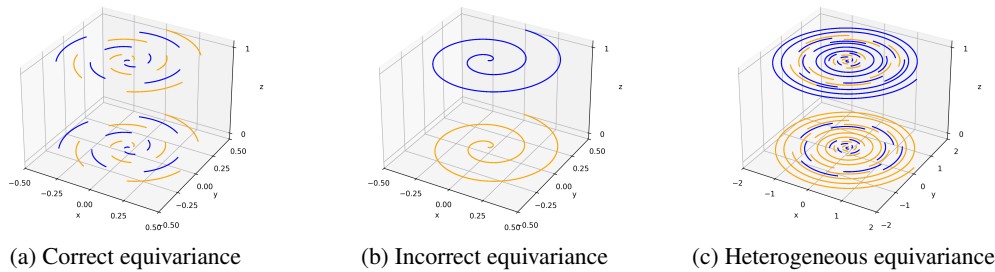

(a) Correct equivariance      (b) Incorrect equivariance      (c) Heterogeneous equivariance

Figure 1: Visualization of the swiss roll datasets. Different color indicates different class labels. (a) Correct equivariance data, (b) Incorrect equivariance data, (c) Heterogeneous equivariance data.

simply the $xy$ plane. Figure 1 shows the visualization of the swiss roll datasets. Following Wang et al. (2024a), we consider data whose label is invariant to reflection over $z$ as correctly equivariant data (Figure 1(a)), and those whose label changes under the reflection as incorrectly equivariant data. (Figure 1(b)).

To better evaluate our models' ability to adapt to heterogeneous symmetry breaking, we extend the separated Swiss Roll dataset into a more challenging variant that exhibits different equivariance types across different regions of the $xy$ plane. We call this dataset heterogeneous Swiss Roll. Specifically, we partition the 2D plane (the quotient space $X/G$) into concentric annular regions centered at the origin, each with a radius width of 0.5. Data within the annuli of radii [0, 0.5] and [1.0, 1.5] exhibit correct equivariance (denoted as Cr1 and Cr2 in the following), while those within [0.5, 1.0] and [1.5, 2.0] exhibit incorrect equivariance (denoted as Icr1 and Icr2 in the following). In this setting, data in each region corresponds to a specific orbitwise equivariance type. The dataset is constructed by uniformly sampling points from each region, ensuring a balanced and structured mixture of correct and incorrect equivariant data. (Figure 1(c))

**Models.**    We compare several models. These include an Invariant Model (IM) (132.8K), an unconstrained MLP (MLP) (133.5K), a Relaxed equivariant model (RM) (132.8K), ARM (136.1K), GRM (135.9K), GARM (136.2K). We also evaluate a set of variants of our models whose relaxed weight generators take the radius as input. We call them ARM (r), GRM (r), and GARM (r). This design leverages the fact that radius on $xy$ plane fully determines the orbitwise equivariance type, enabling the model to adapt its symmetry behavior more effectively.

In this experiment, we use 3-layer MLP architecture as the backbone. For invariant model (IM), to lift the input to the group space, we apply the same linear transformation to the original input and the flipping version of the input, and then concatenate them together. Then, we use an equivariant MLP layers to process the lifted feature. To make the model invariant to flipping the $z$ axis, we use a max pooling act on flip dimension to pool the features after two hidden layers. Then we use a linear layer to map the pooled feature to the output logits. The hidden dimension is set to 256, and the output dimension is set to be 2. For the Relaxed Model (RM), in the lift layer, we add two relaxed weights to scale the feature of input and the flipping version of the input, respectively. Then, we apply relaxed MLP layers formed as equation 3 to process the lifted feature. For the DAREN models, we use a two layer unconstrained MLP with 64 hidden dimension as the relaxed weight generator **w**. We use **e3nn.math.soft_one_hot_linspace** function to expand the dimension of input $(x, y)$ and $r = \text{Norm}(x, y)$ (radius-based relaxed weight models). For the unconstrained MLP, the hidden dimension is set to 362 to keep approximately the same parameter count as the compared models.

**Training.**    All models are trained using the Adam optimizer with a learning rate of 0.001 and a batch size of 100. The training process is configured with a minimum of 50 epochs and a maximum of 10,000 epochs, with early stopping applied if the validation loss does not improve for 1,000 consecutive epochs. The dataset consists of 800 training samples, 400 validation samples, and 1,200 test samples. During training, we monitor validation accuracy after each epoch and save the best model based on validation performance. Test accuracy and equivariance error are evaluated on the best model selected by validation performance. To ensure statistical significance, all experiments are

repeated 20 times with different random seeds, and we report the mean and standard deviation of the results across runs.

**Results.**    We evaluate the performance of different models on the heterogeneous Swiss Roll dataset in terms of test accuracy and equivariance error (EE) across different regions of the domain. We report the test accuracy and EE discrepancy (the error between model EE and ground truth EE) of different models on different regions in Table 1 and Table 2, respectively. All experiments are run with 10 random seeds and the results are reported as mean ± standard deviation.

As expected, IM achieves high accuracy on Cr1 and Cr2 regions, but fails on Icr1 and Icr2 regions due to its rigid symmetry constraint. The results shows that RM and MLP perform worse on the Cr1 and Cr2 regions, also corresponding to their larger EE discrepancy at these regions. This supports our hypothesis that RM inherently face a tradeoff when trained on heterogeneous symmetry data. The radius-based relaxed weight models (ARM(r), GRM(r), GARM(r)) can further improves the performance of the model on Cr1 and Cr2 regions, while maintaining good performance on Icr1 and Icr2 regions.

|          | Overall    | Cr1        | Icr1       | Cr2        | Icr2       |
|----------|------------|------------|------------|------------|------------|
| IM       | 71.9±1.7   | 95.9±2.2   | 49.7±3.8   | 92.0±4.0   | 50.4±4.0   |
| MLP      | 93.6±0.71  | 92.3±2.5   | 98.9±1.0   | 82.4±3.3   | 99.0±0.91  |
| RM       | 94.6±0.82  | 94.2±2.7   | 99.3±0.64  | 85.6±2.6   | 99.1±0.78  |
| ARM      | 96.4±1.1   | **98.2±1.6** | 99.5±0.32 | 89.2±3.4   | 98.9±1.0   |
| GRM      | 94.8±1.1   | 96.0±2.4   | 99.7±0.21  | 84.9±4.1   | 98.8±1.1   |
| GARM     | 96.2±0.91  | **98.2±1.4** | 98.9±0.93 | 88.9±2.7   | 98.8±1.1   |
| ARM(r)   | **99.0±0.48** | 97.7±1.4 | 99.8±0.17  | **98.6±1.2** | 99.8±0.18 |
| GRM(r)   | 97.7±0.89  | 96.8±2.3   | 99.7±0.28  | 94.3±2.4   | 99.8±0.17  |
| GARM(r)  | 98.9±0.22  | 97.9±1.2   | **99.8±0.19** | 98.1±1.0 | **99.9±0.005** |

Table 1: Test accuracy of different models on overall and different region of the heterogeneous Swiss Roll dataset. The best results are highlighted in bold.

|          | Cr1          | Icr1          | Cr2          | Icr2          |
|----------|--------------|---------------|--------------|---------------|
| IM(GD)   | 0.0          | 1.0           | 0.0          | 1.0           |
| MLP      | 0.12±0.043   | 0.022±0.015   | 0.26±0.036   | 0.017±0.015   |
| RM       | 0.091±0.044  | 0.014±0.013   | 0.23±0.045   | 0.013±0.009   |
| ARM      | **0.021±0.020** | 0.011±0.009 | 0.16±0.05    | 0.019±0.017   |
| GRM      | 0.051±0.0340 | 0.008±0.007   | 0.23±0.07    | 0.026±0.022   |
| GARM     | 0.024±0.023  | 0.020±0.010   | 0.15±0.046   | 0.021±0.019   |
| ARM(r)   | 0.027±0.022  | 0.004±0.003   | **0.022±0.02** | 0.0031±0.002 |
| GRM(r)   | 0.032±0.027  | **0.002±0.001** | 0.044±0.032 | 0.0022±0.002 |
| GARM(r)  | 0.027±0.026  | 0.004±0.003   | 0.025±0.02   | **0.002±0.0019** |

Table 2: The EE discrepancy of different models on different region of the heterogeneous Swiss Roll dataset. Note that the EE of IM is always 0, so the EE discrepancy of IM is also the groud truth EE (short for GD). The best results are highlighted in bold.

## 5.2    ANGLE PREDICTION

**Dataset.**    We construct a synthetic dataset to evaluate model performance under controlled symmetry breaking in a 2D vector-based prediction task. Each data sample is a 4D vector $[\mathbf{u}^\top, \mathbf{v}^\top]^\top$ that is concatenated from two vectors $\mathbf{u}, \mathbf{v} \in \mathbb{R}^2$. The label is the angle of the vector sum $\mathbf{u} + \mathbf{v}$, discretized into $N$ angular bins in $[0, 2\pi)$. To be specific, let $\Theta(\cdot)$ be the function that maps a 2D vector to its angle, and $BIN_N(\cdot)$ be the function that maps an angle to one of the $N$ bins, i.e., $BIN_N(\theta) = \lfloor \frac{N}{2\pi}\theta \rfloor$ mod $N$. The label can be written as $BIN_N(\Theta(\mathbf{u} + \mathbf{v}))$. It is easy to see that the task is exactly equivariant to $C_N$ group that act on each 2D vector by rotation.

To introduce orbitwise variation in symmetry structure, we uniformly divide the radial range $[0, 2]$ in the $\mathbf{u}$ vector space into five concentric annuli, each associated with a different vertical translation offset $t \in \{0.0, 0.1, 1.0, 2.0, 5.0\}$ add on the sum, i.e. the label become $BIN_N(\Theta(\mathbf{u} + \mathbf{v} + [0, t]^\top))$. When $t = 0$, the task is exactly rotation equivariant; as $t$ increases, the equivariance is progressively broken due to the fixed vertical bias added to the vector sum $\mathbf{u} + \mathbf{v}$. The vector $\mathbf{u}$ and $\mathbf{v}$ are uniformly sampled in the polar coordinate space, with radius range $[0, 2]$ and angle range $[0, 2\pi]$. In this experiment, we set the number of bins $N = 16$.

**Model.** We compare an Equivariant Model (EM) (70.1K), an MLP (71.2K), RM (70.1K) ARM (71.6K), GRM (70.6K), and GARM (71.7K). All models are based on 3-layer MLP architecture. For EM, we use $C_{16}$-equivariant MLP layers as the backbone, with hidden dimension 64 and output dimension 16. (Also, the first layer is a lifting layer that lifts the input to group space.) Then we design RM based on EM by replacing the equivariant MLP layers with relaxed equivariant MLP layers. For DAREN model, we use a two layer unconstrained MLP with 32 hidden dimension as the relaxed weight generator $\mathbf{w}$ and take radius of $\mathbf{u}$ as its' input. We also use **e3nn.math.soft_one_hot_linspace** function to process the input. For the MLP, the hidden dimension is set to 256 to keep approximately the same parameter count as the compared models.

**Training.** The training setup is similar to the Swiss Rolls experiment, except that the dataset consists of 2,000 training samples, 200 validation samples, and 800 test samples. Besides, we also add an equivariant constraint loss with a weight of 1.0 to the training objective for all models except EM and MLP. Same as Wang et al. (2022), the equivariant constraint loss is computed as the sum of pairwise differences between the relaxed weights within each layer.

**Results.** The results of test accuracy and EE discrepancy across different $t$ values are reported in Table 3 and Table 4, respectively. As expected, EM achieves the highest accuracy at $t = 0$, but its performance degrades significantly as $t$ increases due to the increasing symmetry breaking. Our proposed models outperform the MLP and RM for most $t$ values, demonstrating their ability to adapt to varying symmetry breaking levels. The gated models (GRM and GARM) outperform their non-gated counterparts (RM and ARM) on most $t$ values, indicating that the gating mechanism can effectively stabilize training and improves robustness when group size increases.

| | Overall | $t=0$ | $t=0.1$ | $t=1.0$ | $t=2.0$ | $t=5.0$ |
|---|---|---|---|---|---|---|
| EM | 43.6±1.8 | **96.7±0.9** | 79.9±4.0 | 20.0±4.6 | 12.1±2.6 | 9.0±2.6 |
| MLP | 78.6±1.7 | 89.2±2.3 | 75.1±5.2 | 68.0±4.0 | 73.3±4.0 | 87.2±2.9 |
| RM | 81.1±1.6 | 93.6±2.2 | 87.9±3.4 | 68.4±4.7 | 72.6±4.4 | 83.0±3.4 |
| ARM | 84.4±1.6 | 94.3±2.5 | 85.0±2.6 | 74.0±3.8 | 78.6±4.2 | **90.1±2.3** |
| GRM | 85.9±1.1 | 96.1±1.1 | **88.8±4.3** | 78.6±3.5 | 78.2±4.9 | 87.9±2.1 |
| GARM | **86.4±1.3** | 95.9±1.3 | 85.2±2.6 | **80.1±2.5** | **81.4±3.7** | 89.4±3.2 |

Table 3: Test accuracy of different models on the angle prediction dataset. The best results are highlighted in bold.

| | $t=0$ | $t=0.1$ | $t=1.0$ | $t=2.0$ | $t=5.0$ |
|---|---|---|---|---|---|
| EM(GD) | 0.00 | 0.5167 | 1.6954 | 1.8672 | 1.965 |
| MLP | 0.32±0.04 | 0.36±0.04 | 0.041±0.02 | 0.013±0.006 | 0.012±0.005 |
| RM | 0.16±0.04 | **0.053±0.03** | 0.046±0.04 | 0.019±0.01 | 0.018±0.007 |
| ARM | 0.13±0.03 | 0.072±0.04 | 0.032±0.02 | 0.014±0.01 | 0.0063±0.005 |
| GRM | **0.063±0.01** | 0.076±0.07 | 0.019±0.01 | 0.024±0.02 | 0.0091±0.0066 |
| GARM | 0.068±0.01 | 0.058±0.04 | **0.018±0.01** | **0.012±0.01** | **0.0055±0.0042** |

Table 4: The EE discrepancy of different models on the angle prediction dataset. Note that the EE of EM is always 0, so the EE discrepancy of EM is also the ground truth EE (short for GD). The best results are highlighted in bold.

## 6    CONCLUSION AND FUTURE WORK

In this work, we propose a data-adaptive model for domains with heterogeneous symmetry breaking. By introducing orbit-conditioned relaxed weights and a learnable gating mechanism, our method can adjust the model's equivariance based on the symmetry breaking level of each orbit. Experiments on synthetic datasets show that our approach outperforms equivariant models, relaxed models and unconstrained models, especially in settings with varying levels of symmetry breaking.

The limitations of this work include the reliance on synthetic datasets for evaluation, which may not fully capture the complexity of real-world data. Future work could explore applications of our method to real-world datasets with heterogeneous symmetry properties, such as molecular data or robotics.

**Acknowledgments**    The authors are grateful to Rui Wang, Jung Yeon Park, Xupeng Zhu and Dian Wang for helpful discussions. R.W. would like to acknowledge support from NSF Grants 2442658 and 2134178. This work is supported by the National Science Foundation under Cooperative Agreement PHY-2019786 (The NSF AI Institute for Artificial Intelligence and Fundamental Interactions, http://iaifi.org/).

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
