# OpenReview forum: "Data-Adaptive Relaxed Equivariant Networks for Symmetry Breaking"
_ICLR.cc/2026/Workshop/GRaM — ICLR 2026 Workshop GRaM Poster_

### Official Review · Reviewer_yQYL · 2026-02-14
**Review of the paper**

**Rating:** 7
**Confidence:** 3

**Review:**

Summary:
The authors introduce a method which condition the degree of equivariance on the input’s orbit in the quotient space and validate the idea empirically on two synthetic benchmarks.

Strengths:
- The paper is generally well written and easy to follow.
- The orbitwise equivariance notion is a useful concept for reasoning about symmetry mismatch at the level of group orbits rather than points or the entire domain.
- I find this paper a good fit for this workshop.

Weaknesses:
- Synthetic tasks are well-designed for controlled tests of heterogeneity; still, adding at least one real-world dataset (e.g., molecular properties with known approximate symmetries) would substantively increase impact.
- Several methodological aspects would benefit from tightening, e.g., guarantees or concrete designs for invariant encoders when X/G is not explicit

**Pmlr Suitability:**

Yes

---

### Official Review · Reviewer_PSbY · 2026-02-23

**Rating:** 7
**Confidence:** 3

**Review:**

Summary: This paper extends relaxed equivariance networks to be able to change the relaxation weights per orbit. They also introduce gating, which allows the network to interpolate between equivariance and non equivariance on each orbit separately. They then test these notions with a couple experiments.

Strengths: The paper is easy to follow and all the definitions are introduced well.
The method is simple and performs well on the tasks.
I think the gating idea is a very interesting avenue of research to explore, so this is a very good paper for this workshop.

Weaknesses: The relaxed equivariance networks operate by learning measure on the symmetry group for each orbit, or equivalently an invariant function from X to the space of measures on G. This is the same as learning a (weighted) frame, but the relevant literature is not cited.

It would be nice to have a sense for some problems where it is known that equivariance doesn’t hold for all orbits, just to see where this method can be applied.

Comment: A small technical comment, but it should be mentioned that G is assumed to be finite, though the gating could be extended to arbitrary groups.

**Pmlr Suitability:**

Yes

---

### Official Review · Reviewer_9Xy2 · 2026-02-24
**Difficult to read, but potentially interesting.**

**Rating:** 4
**Confidence:** 3

**Review:**

This paper introduces Data-Adaptive Relaxed Equivariant Networks (DAREN) to handle orbit-based symmetry breaking across different data orbits. Overall this appears to be a small perturbation from existing work, effectively considering the problem setting where the group action is orbit dependent: in particular, I am unclear whether this falls under the umbrella of what people normally refer to as symmetry breaking: effectively the problem is no longer f(x) = g^-1f(gx), but f(x) = action(g^-1, Gx). f(gx), so the overall problem is just different? But most importantly, the symmetry is still present.

This issue particularly manifests as all the examples considered in this paper are very specific synthetic toy examples, with no clear real world problems, of why such a setting may be of interest -- adding this would be of significant interest. Without it, this paper introducing a fragmented array of model variations makes it difficult to determine if the added complexity of gating mechanisms and orbit conditioning is genuinely needed for broader tasks.

In addition to this, the authors do not specify what the last layer is meant to look like -- from a point of view of the problem no longer being an invariance problem, but the action being orbit dependent, this can easily be captured by an appropriate projection layer at the end.

Furthermore, I find that the related literature is extremely limited: various works on frames and canonicalization should be mentioned, once the problem becomes orbit - dependent.

**Pmlr Suitability:**

No

---

### Meta-Review · Area_Chair_WkXL · 2026-02-25

**Decision:**

Accept

**Metareview:**

For the most part, the reviewers say that the paper is well-written, and find the methods and ideas introduced to be interesting. Moreover, they find the paper to be a good fit for the workshop. Weaknesses are that the experiments are toy and it is not clear for which real world tasks the method would be applicable. Nonetheless, the paper seems correct with a novel extension that could be of interest to the community so I recommend acceptance.

**Relevance To Proceedings:**

Yes — suitable for PMLR (long paper)

**Relevance To Workshop:**

Yes — suitable for GRaM

---

### Decision · Program_Chairs · 2026-03-02

Accept (Poster)